# Incidence and Risk Factors of Refeeding Syndrome in Preterm Infants

**DOI:** 10.3390/nu16152557

**Published:** 2024-08-03

**Authors:** Suzan S. Asfour, Belal Alshaikh, Maya Mathew, Dina I. Fouda, Mountasser M. Al-Mouqdad

**Affiliations:** 1Clinical Pharmacy Department, Pharmaceutical Care Services, King Saud Medical City, Riyadh 12746, Saudi Arabia; asfsuzan@gmail.com (S.S.A.);; 2Department of Pediatrics, Cumming School of Medicine, University of Calgary, Calgary, AB T2N 1N4, Canada; 3Neonatal Intensive Care, Hospital of Pediatrics, King Saud Medical City, Riyadh 12746, Saudi Arabia

**Keywords:** parenteral nutrition, refeeding syndrome, sodium phosphate, preterm infants

## Abstract

This study aimed to evaluate the incidence and risk factors associated with refeeding syndrome (RFS) in preterm infants (≤32 weeks gestational age) during their first week of life. Infants (gestational age ≤ 32 weeks; birth weight < 1500 g) who were admitted to the neonatal intensive care unit (NICU), level III, and received parenteral nutrition between January 2015 and April 2024 were retrospectively evaluated. Modified log-Poisson regression with generalized linear models and a robust variance estimator was applied to adjust the relative risk of risk factors. Of the 760 infants identified, 289 (38%) developed RFS. In the multivariable regression analysis, male, intraventricular hemorrhage (IVH), and sodium phosphate significantly affected RFS. Male infants had significantly increased RFS risk (aRR1.31; 95% CI 1.08–1.59). The RFS risk was significantly higher in infants with IVH (aRR 1.71; 95% CI 1.27–2.13). However, infants who received higher sodium phosphate in their first week of life had significantly lower RFS risk (aRR 0.67; 95% 0.47–0.98). This study revealed a notable incidence of RFS among preterm infants aged ≤32 gestational weeks, with sex, IVH, and low sodium phosphate as significant risk factors. Refined RFS diagnostic criteria and targeted interventions are needed for optimal management.

## 1. Introduction

Despite the lack of a universally accepted definition of refeeding syndrome (RFS), most studies agree that it is characterized by fluid and electrolyte imbalances occurring when nutrition is reintroduced after prolonged malnutrition or starvation [1,2]. RFS can result from both enteral nutrition and parenteral nutrition (PN), and it often involves significant electrolyte disturbances, particularly phosphorus, and can lead to severe complications affecting multiple organ systems, potentially resulting in death [1,2,3]. 

Studies have found that adult patients with conditions including tuberculosis, cancer, psychiatric disorders, and chronic diseases who are malnourished are at risk of developing RFS [4,5]. Consequently, these patients require a longer hospital stay and have a higher mortality risk during hospitalization [6,7,8]. RFS was first recognized in adult patients who experienced severe malnutrition or starvation, such as prisoners of war, individuals with anorexia nervosa, or those recovering from famine [4,5,9]. Upon refeeding, these individuals developed potentially life-threatening symptoms. The underlying mechanism involves a metabolic shift from a catabolic state, where the body primarily uses fat and protein for energy, to an anabolic state, where carbohydrate intake stimulates insulin release. Insulin promotes the cellular uptake of glucose, potassium, magnesium, and phosphate. This sudden shift can lead to profound hypophosphatemia, as phosphate is rapidly utilized in cells for ATP production and other metabolic processes [10]. The resultant electrolyte imbalance, particularly low phosphate levels, can cause severe complications, including cardiac, respiratory, and neurological dysfunction [6,7,8]. RFS is also observed in the pediatric population, particularly among those admitted to a pediatric intensive care unit (PICU) [11]. Up to 25% of children admitted to a PICU are malnourished, placing them at a heightened risk of RFS development if nutrition is delayed for >5 days [11,12,13]. A study reported a mortality rate of up to 6% among children with RFS [14].

Neonates were believed to be not at risk of RFS because they received enteral nutrition or PN shortly after birth. However, neonates, particularly those who are extremely premature, small for gestational age (SGA), affected by intrauterine growth restriction (IUGR), or have very low birth weight (VLBW), can become malnourished at a very early stage of life, putting them at risk for RFS [2]. 

The incidence of RFS in the neonatal population varies widely because of inconsistencies in its definition. This variability stems from whether the diagnosis is based solely on hypophosphatemia or includes hypokalemia and hypomagnesemia. The varied definition of hypophosphatemia and differences in inclusion criteria across studies further contributed to the wide range in the reported incidence rates [15,16,17,18,19,20]. 

Several studies have identified fetal growth restriction, irrespective of its cause—placental insufficiency or genetic disorders—as a significant risk factor for RFS development in newborns [15,19]. Other studies have highlighted that being SGA is also associated with an increased RFS risk [16,19,20]. Furthermore, reports have suggested that the early introduction of high amino acid content through PN in premature infants carries a substantial risk of RFS [16,21].

Despite these findings, these studies share several limitations. They often have small sample sizes, include both preterm and term infants, and utilize varying definitions of RFS and hypophosphatemia [17,18,19,22]. Moreover, many of these studies do not report the concentrations of other PN components, such as dextrose and proteins [15,17,23]. 

These factors make it challenging to ascertain the true incidence rate and risk factors of RFS in preterm babies. Therefore, this study aimed to determine the incidence and risk factors for RFS development in premature infants aged ≤32 gestational weeks during the first week of life.

## 2. Materials and Methods

### 2.1. Study Design

This retrospective chart review included preterm infants who were admitted to the neonatal intensive care unit (NICU) of the King Saud Medical City (KSMC), a tertiary referral center, between January 2015 and June 2024. The NICU, level III, at KSMC has an average annual admission of 1100 patients. 

This study was conducted in accordance with the Declaration of Helsinki and Good Pharmacoepidemiology Practice Guidelines and was approved by the Medical Ethical Review Committee of KSMC (Reference number H1RI-12-May24-01), which also waived the need for informed consent.

### 2.2. Inclusion and Exclusion Criteria

Very preterm infants who weighed <1500 g at birth, were born at KSMC at ≤32 weeks of gestation, and were admitted to the NICU, level III, were included. All the included newborns received PN plus lipid emulsion within the first 24 h of life. Infants with a known chromosomal or genetic abnormality and significant congenital defects or congenital infections, did not receive PN, were not born at KSMC or were transferred to another hospital or died within the first 7 days, and had nonretrievable data were excluded from the analysis.

### 2.3. Data Collection and Follow-Up

Neonatal data from NICU admission until discharge or death were retrieved. Demographic data and clinical and outcome data, including major morbidities associated with prematurity, were also reviewed and collected. Maternal data, including antenatal steroid treatment, mode of delivery, and presence of gestational diabetes mellitus and maternal hypertension, were also obtained. 

### 2.4. Study Outcome

The primary outcomes of this study were the incidence of RFS in the first 7 days after birth and risk factors of RFS in very preterm infants.

### 2.5. Definitions

#### Nutrition Protocol

PN: PN was started early after birth using starter PN. Individualized PN was prescribed daily. Starter PN contains 10% dextrose, 4% amino acids, and 0.01 mmol/mL calcium gluconate [2]. Individualized PN solution containing amino acids (3.5–4 g/kg/day), dextrose (5–12 mg/kg/min), lipid emulsion (1–3 g/kg/day), sodium chloride (1–3 mmol/kg/day), sodium acetate (1–2 mmol/kg/day), sodium phosphate (1–2 mmol/kg/day), potassium chloride (1–3 mmol/kg/day), potassium acetate (1–2 mmol/kg/day), potassium phosphate (1–2 mmol/kg/day), trace elements (Peditrace^®^), and water- and fat-soluble vitamins (Soluvit^®^ N and Vitalipid^®^ N Infant, respectively) was started within the first 24 h of life and infused continuously for 24 h [2].

RFS: A clear definition of neonatal RFS has not yet been established: hypercalcemia (>2.8 mmol.L^−1^), hypophosphatemia (between >1.1 and <1.6 mmol.L^−1^), and severe hypophosphatemia (<1.0 mmol.L^−1^) [16,22,24,25,26]. 

### 2.6. Statistical Analysis

Before the analysis, the dataset was reviewed and checked for missing data. Data were analyzed using IBM SPSS Statistics for Windows version 25.0 (IBM Corp., Armonk, NY, USA).

Infant and maternal variables were presented using descriptive statistics, including median, interquartile range (IQR), frequency, and percentage. The Mann–Whitney U test was used for between-group comparisons of ordinal qualitative variables. Fisher’s exact test was utilized to determine the association between categorical variables. For between-group comparisons of continuous variables, the unpaired Student’s *t*-test was used for normally distributed data, whereas the Mann–Whitney U test was used for non-normally distributed data. The Kolmogorov–Smirnov test and a visual inspection of histograms were performed to evaluate the distribution of quantitative variables.

To determine the risk factors of RFS in premature infants, a univariate relative risk analysis on the recorded variables (gestational age, birth weight, SGA, delivery mode, sex, 1 and 5 min Apgar scores, maternal hypertension, antenatal steroid treatment, premature rupture of the membrane, gestational diabetes mellitus, necrotizing enterocolitis, surfactant use, late-onset sepsis, dextrose intake, amino acids, lipid emulsion, sodium chloride, sodium acetate, sodium phosphate, potassium chloride, potassium acetate, potassium phosphate, magnesium sulfate, calcium gluconate, and calcium-to-phosphate ratio) was first performed because they were considered potential confounders. All factors with a *p*-value of <0.05 in the univariate analysis were included in the final multivariable regression model. Modified log-Poisson regression with generalized linear models and a robust variance estimator (Huber–White) were applied for univariate relative risk analysis and to the models to adjust the relative risk for RFS risk factors in premature infants. All statistical tests were two-tailed, and *p*-values of <0.05 were considered significant.

## 3. Results

During the study period, 2465 preterm infants with ≤32 gestational weeks and birth weight < 1500 g were admitted to the level 3 NICU. Of them, 760 met the inclusion criteria and were eligible for inclusion in the final analysis (Figure 1). 

Among the 760 infants, RFS developed in 289 (38%). Maternal and neonatal characteristics are summarized in Table 1. Figure 2 shows the percentages of RFS, hypercalcemia, hypophosphatemia, and severe hypophosphatemia. Infants with RFS have lower gestational age, birthweight, and length and head circumference than infants without RFS (*p* < 0.001). The Apgar scores at 1 and 5 min were also lower in infants with RFS than in those without RFS (*p* < 0.001). In addition, male infants were at higher risk of RFS than female infants (*p* < 0.001). Infants with RFS received more surfactant and required mechanical ventilators more often than those without RFS (*p* < 0.001). Moreover, infants with RFS had more severe IVH than infants without RFS (*p* < 0.001).

In the univariate analysis, the median intake of macronutrients, including parenteral lipids (g/kg/day) and amino acids (g/kg/day), in the first postnatal week was significantly higher in infants with RFS than in those without RFS (*p* < 0.001). TPN duration was also significantly higher in infants with RFS than in those without it (*p* = 0.002) (Table 2).

Interestingly, the median intake of sodium chloride, sodium phosphate, and potassium chloride in the first postnatal week was significantly higher in infants without RFS than in those with RFS (*p* < 0.001, <0.001, 0.001, respectively) (Table 2). 

In contrast, the median intake of calcium gluconate and potassium acetate and the calcium-to-phosphate ratio in the first postnatal week were significantly higher in infants with RFS than in those without RFS (*p* = 0.001, <0.001, <0.001, respectively) (Table 2).

The multivariable regression analysis performed after adjusting the variables that were significant in the univariate analysis revealed that male sex, IVH, and sodium phosphate significantly affect RFS risk. Male infants had significantly increased RFS risk (aRR1.31; 95% CI1.08–1.59). The RFS risk was significantly higher in infants with and without IVH (aRR 1.71; 95% CI 1.27–2.13). However, infants who received higher sodium phosphate in their first week of life had a significantly lower RFS risk (aRR 0.67; 95% 0.47–0.98) (Table 3).

Furthermore, the risk factors of RFS in neonates with gestational age < 28 weeks were analyzed. Among the 264 infants, RFS developed in 136 (51.5%). Infants with RFS had lower gestational age, birthweight, head circumference, and Apgar score at 1 min than infants without RFS (*p* = 0.02, 0.008, 0.03, 0.03, respectively) (Table 1).

Infants aged <28 gestational weeks with RFS received lower amounts of dextrose, sodium acetate, sodium phosphate, and potassium chloride than infants without RFS (*p* = 0.03, 0.02, 0.001, 0.001, respectively) (Table 2). In addition, the average calcium-to-phosphate ratio intake in the first week of life was higher in infants with RFS than in those without RFS (*p* < 0.001) (Table 2).

The multivariable regression analysis after adjusting the confounders revealed that infants who received higher amounts of sodium phosphate in the first week of life had a significantly lower RFS risk (aRR 0.60; 95% CI 0.38–0.96) (Table 3).

## 4. Discussion

This study examined the incidence rate and identified the risk factors associated with RFS in preterm infants. The findings revealed that the incidence of RFS was approximately 38%, highlighting the significant prevalence of this condition in this study population.

The incidence rates of RFS in the neonatal population vary widely, ranging from 20% to 90%. This variability can be attributed to several factors, including differences in the definition of RFS across studies. Some studies define RFS solely based on hypophosphatemia, whereas others include additional electrolyte imbalances such as hypokalemia or hypercalcemia [16,17,18,20,21,23,27]. In addition, the incidence rates are influenced by the prevalence of IUGR and SGA within the studied populations and the amounts of amino acids provided in PN during the first few days of life.

In this study, RFS was defined based on the ProVIDe Trial, which defines serum hypophosphatemia as <1.4 mmol/L and hypercalcemia as adjusted calcium >2.8 mmol/L [20,28]. The use of this standardized definition allows for more accurate comparisons of our findings with those of other studies and emphasizes the need for consistent criteria in evaluating the incidence and risk factors of RFS.

In this study, several key risk factors were found to be associated with RFS development in preterm infants during the first week of life. The findings indicate that sex, IVH, and lower sodium phosphate intake in the first week of life are significant risk factors for RFS. 

Male infants were found to have a higher risk of RFS development than female infants. This finding is particularly intriguing because sex has not been widely recognized as a risk factor for RFS in the existing literature. The ProVIDe Trial is one of the few studies suggesting a potential link, noting a trend toward increased RFS incidence in male infants without definitively establishing sex as a risk factor [20]. Our findings support this observation, indicating that being of male sex may contribute to RFS susceptibility in preterm infants. The increased risk in male infants may be due to hormonal or genetic factors, although this hypothesis requires further investigation.

This study found a significant association between IVH and RFS development in preterm infants during their first week of life. This finding aligns with two other studies that reported a significant association between IVH and hypophosphatemia, a key component of RFS [20,23]. However, four other studies did not find a significant association [15,21,29,30]. This discrepancy in the findings could be attributed to differences in the overall incidence of IVH in study populations, sample sizes, and clinical protocols. The highest incidence of IVH in very premature infants typically occurs within the first week of life, coinciding with the period when RFS is most likely to develop. Given that IVH is a severe morbidity associated with prematurity, the significant association with RFS highlights the need for close monitoring and management of these infants during this critical period. 

This study identified low sodium phosphate intake as a significant risk factor for RFS development in preterm infants, particularly those aged <28 weeks of gestation. Notably, our institution does not initiate sodium phosphate in PN within the first 2 days of life. Consequently, a lower incidence of RFS was noted in preterm infants who received higher amounts of phosphate over the first 7 days of life. These findings are consistent with those of Bustos-Lozano et al. and Mulla et al. [27,31]. Bustos-Lozano et al. reported hypophosphatemia in 29% of infants who received phosphate within their first 48 h of life compared with 69% in those who did not receive early phosphate supplementation. Similarly, Mulla et al. found hypophosphatemia in 60% of infants with a calcium-to-phosphate ratio of 1.3–1.5:1 compared with 35% in those with a 1:1 ratio. These results highlight the importance of early phosphate supplementation and adjusting the calcium-to-phosphate ratio to reduce the risk of RFS and its complications. 

High amino acid intake was not identified as a risk factor for RFS development in our cohort. Our neonatal unit adheres to the recommendation of providing adequate protein immediately after birth, administering 3.5–4 g/kg/day to support growth and neurodevelopmental outcomes [32,33]. As a result, infants with and without RFS received the same amount of enteral and parenteral amino acids. We do not practice adjusting the rate of amino acid intake based on serum urea levels because this measure is not considered reliable in preterm infants. 

This study has several limitations. First, the retrospective design and single tertiary center setting may limit the generalizability of the findings. Second, the association between IUGR and RFS could not be investigated because of a lack of antenatal care information for the majority of our infants’ mothers, as many did not receive antenatal care or we did not have access to their antenatal care history.

Despite these limitations, this study has several strengths. The definition of RFS used in the ProVIDe Trial, a large multicenter prospective study with a substantial sample size, was adopted, which enhances the reliability and comparability of our results. Furthermore, this study has a robust sample size compared with many previous studies, allowing for more reliable statistical analysis. Moreover, we meticulously tracked changes in the components of PN between infants with and without RFS. These components included proteins, carbohydrates, and lipids, as well as micronutrients, such as magnesium and sodium and potassium supplements.

## 5. Conclusions

In our NICU population, the incidence of RFS is high, revealing significant clinical concerns. Some risk factors, such as sex, are beyond our control, whereas others, including nutritional protocols and strategies to minimize the risk of IVH, can be managed more effectively. This study highlights the necessity for further advanced prospective studies to accurately define neonatal RFS and determine optimal amino acid intake and the early initiation of phosphate supplementation. Although our findings contribute to the growing body of evidence, more studies are essential to develop more effective prevention and management strategies for preterm infants at risk of RFS, ultimately improving their outcomes and care.

## Figures and Tables

**Figure 1 nutrients-16-02557-f001:**
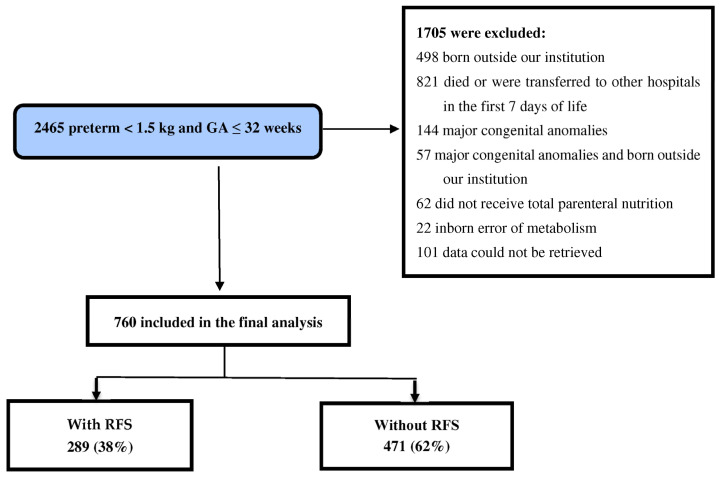
Flow chart of patient selection. GA: gestational age.

**Figure 2 nutrients-16-02557-f002:**
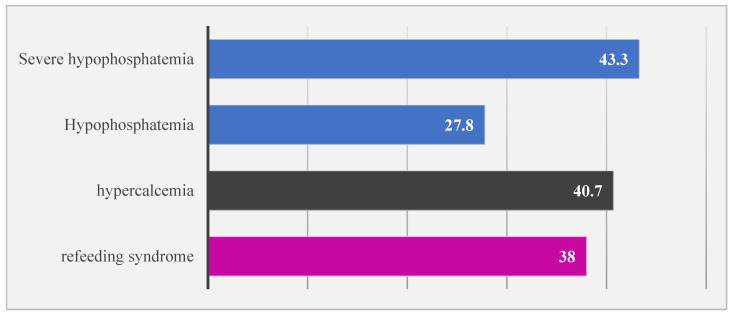
Percentages of refeeding syndrome, hypercalcemia, hypophosphatemia, and severe hypophosphatemia (*n* = 760).

**Table 1 nutrients-16-02557-t001:** Demographic characteristics of the mothers and infants with or without refeeding syndrome (RFS, *n* = 760).

Variable	ALL				Gestational Age < 28 Weeks			
	N	without RFS (*n* = 471)	with RFS (*n* = 289)	*p*-Value	N	without RFS (*n* = 128)	with RFS (*n* = 136)	*p*-Value
Gestational age (weeks), median (IQR)	760	29 (27.0–31.0)	28 (26–30.0)	**<0.001 ***	264	26 (25.0–27.0)	26 (25.0–27.0)	**0.02 ***
Birth weight (grams) (IQR)	760	1180 (950–1370)	950 (765–1200)	**<0.001 ***	264	845 (711.25–960)	775 (661.25–905)	**0.008 ***
Length (cm), median (IQR)	760	38 (35–40)	35 (32–38)	**<0.001 ***	264	33 (31–35)	33 (31–35)	0.10
Head circumference (cm) (IQR)	760	27 (25–28)	25 (23–27)	**<0.001 ***	264	24 (23–25)	23 (22–25)	**0.03 ***
Small for gestational age, *n* (%)	760	63 (13.4)	47 (16.3)	0.29	264	8 (6.3)	9 (6.6)	1
1 min Apgar score, median (IQR)	760	6 (4–7)	5 (3–6)	**<0.001 ***	264	5 (3–6)	4 (2–6)	**0.03 ***
5 min Apgar score, median (IQR)	760	7 (6–8)	7 (7–8)	**<0.001 ***	264	5 (6–7)	4 (6–7)	0.17
Male, *n* (%)	760	221 (46.9)	174 (60.2)	**<0.001 ***	264	70 (54.7)	90 (66.2)	0.06
Antenatal steroid treatment, *n* (%)	760	253 (53.7)	147 (50.9)	0.45	264	70 (54.7)	71 (52.2)	0.71
Gestational diabetes mellitus, *n* (%)	760	27 (5.7)	14 (4.8)	0.37	264	7 (5.5)	4 (2.9)	0.36
Maternal hypertension, *n* (%)	760	119 (25.3)	67 (23.2)	0.54	264	26 (20.3)	27 (19.9)	1.0
Preterm rupture of membrane, *n* (%)	760	59 (12.5)	20 (6.9)	**0.01 ***	264	17 (13.3)	12 (8.8)	0.33
Cesarean section, *n* (%)	760	223 (47.3)	146 (50.5)	0.41	264	72 (56.3)	75 (55.1)	0.90
Expressed breast milk, *n* (%)	760	247 (52.4)	145 (50.2)	0.55	264	75 (58.6)	64 (47.1)	0.06
Respiratory distress syndrome required surfactant, *n* (%)	760	281 (59.7)	224 (77.5)	**<0.001 ***	264	120 (93.8)	128 (94.1)	1
Mechanical ventilation, *n* (%)	760	305 (64.8)	242 (83.7)	**<0.001 ***	264	122 (95.3)	132 (97.1)	0.53
Patent ductus arteriosus requiring treatment, *n* (%)	760	35 (7.4)	34 (11.8)	0.05	264	24 (18.8)	28 (20.6)	0.76
Intraventricular hemorrhage, *n* (%)	760	124 (26.3)	157 (54.5)	**<0.001 ***	264	64 (50)	96 (70.6)	**0.001 ***
Intraventricular hemorrhage grades 3 and 4, *n* (%)	760	63 (13.3)	74 (25.6)	**<0.001 ***	264	41 (32)	56 (41.2)	0.13

* *p*-values < 0.05.

**Table 2 nutrients-16-02557-t002:** Nutritional and electrolyte supplementation characteristics of infants with or without refeeding syndrome (RFS, *n* = 760).

Variables	ALL				Gestational Age < 28 Weeks			
	N	without RFS (*n* = 471)	with RFS (*n* = 289)	*p*-Value	N	without RFS (*n* = 128)	with RFS (*n* = 136)	*p*-Value
Average parenteral lipid intake in the 1st 7 days (g/kg/day), median (IQR)	760	2 (1.54–2.35)	2.1 (1.67–2.5)	**0.02 ***	264	1.8 (1.4–2.3)	1.8 (1.3–2.4)	0.95
Average parenteral amino acid intake in the 1st 7 days (g/kg/day), median (IQR)	760	3.90 (3.67–4.0)	3.97 (3.75–4.0)	**0.01 ***	264	4 (3.7–4.0)	4 (3.8–4.0)	0.13
Average parenteral dextrose intake in the 1st 7 days (mg/kg/min), median (IQR)	760	8.47 (7.74–9.12)	8.46 (7.65–9.04)	0.55	264	8 (7–8.8)	7.7 (6.4–8.5)	**0.03 ***
Average parenteral phosphate intake in the 1st 7 days (mg/kg/min), median (IQR)	760	0.30 (0–0.6)	0.24 (0–0.4)	**0.01 ***	264	0.30 (0.02–0.62)	0.2 (0–0.44)	**0.03 ***
PN duration, median (IQR)	760	14 (7–29)	18 (9–35)	**0.002 ***	264	28 (12–48)	23 (11–41)	0.45
Average magnesium sulfate intake in the 1st 7 days (mmol/kg/day), median (IQR)	760	0.11 (0.07–0.13)	0.11 (0.07–0.13)	0.60	264	0.10 (0.07–0.13)	0.11 (0.07–0.14)	0.24
Average calcium gluconate intake in the 1st 7 days (mmol/kg/day), median (IQR)	760	0.71 (0.57–0.92)	0.79 (0.61–1)	**0.001 ***	264	0.77 (0.64–0.96)	0.86 (0.64–1)	0.15
Average sodium chloride intake in the 1st 7 days (mmol/kg/day), median (IQR)	760	0.14 (0–0.60)	0 (0–0.40)	**<0.001 ***	264	0.0 (0.0–0.43)	0.0 (0.0–0.29)	0.08
Average sodium acetate intake in the 1st 7 days (mmol/kg/day), median (IQR)	760	0.57 (0–1.14)	0.43 (0–0.96)	0.08	264	0.86 (0.104–1.14)	0.57 (0.0–1.0)	**0.02 ***
Average sodium Phosphate intake in the 1st 7 days (mmol/kg/day), median (IQR)	760	0.37 (0.14–0.60)	0.29 (0–0.50)	**<0.001 ***	264	0.3 (0.1–0.59)	0.2 (0.0–0.43)	**0.001 ***
Average potassium chloride intake in the 1st 7 days (mmol/kg/day), median (IQR)	760	0.4 (0.0–1.0)	0.2 (0.0–0.8)	**0.001 ***	264	0.4 (0.0–1.0)	0.0 (0.0–0.6)	**0.001 ***
Average potassium acetate intake in the 1st 7 days (mmol/kg/day), median (IQR)	760	0 (0–0.43)	0.14 (0–0.71)	**<0.001 ***	264	0.14 (0–0.64)	0.29 (0–0.86)	0.11
Average potassium phosphate intake in the 1st 7 days (mmol/kg/day), median (IQR)	760	0 (0–0.38)	0.1 (0.0–0.38)	0.14	264	0.0 (0.0–0.42)	0.1 (0.0–0.40)	0.14
Average calcium-to-phosphate ratio intake in the 1st 7 days (mmol/kg/day), median (IQR)	760	0.61 (0.38–0.90)	0.87 (0.54–1.44)	**<0.001 ***	264	0.65 (0.42–0.99)	0.97 (0.69–1.68)	**<0.001 ***

* *p*-values < 0.05.

**Table 3 nutrients-16-02557-t003:** Univariate and multivariable regression analyses of risk factors of refeeding syndrome (RFS) in very preterm infants.

Risk Factor	All				Gestational Age < 28 Weeks			
	Unadjusted RR 95% CI	*p*-Value	Adjusted RR 95% CI	*p*-Value	Unadjusted RR 95% CI	*p*-Value	Adjusted RR 95% CI	*p*-Value
Gestational age	0.89 (0.86–0.92)	**<0.001 ***	1.02 (0.95–1.09)	0.59	0.90 (0.83–0.99)	**0.03 ***	1.0 (0.89–1.13)	0.98
Birth weight	0.99 (0.98–1.0)	**<0.001 ***	0.99 (0.98–1.0)	0.07	0.99 (0.98–1)	**0.01 ***	1.0 (0.99–1.001)	0.63
Small for gestational age	1.15 (0.90–1.46)	0.25	–	–	0.97 (0.61–1.55)	0.90	–	–
Male	1.40 (1.16–1.69)	**<0.001 ***	1.31 (1.08–1.59)	**0.006 ***	0.78 (0.61–1.01)	0.06	–	–
Cesarean section	1.08 (0.90–1.30)	0.40	–	–	1.02 (0.81–1.29)	0.86	–	–
1 min Apgar score	0.89 (0.85–0.93)	**<0.001 ***	0.98 (0.91–1.04)	0.48	0.93 (0.88–0.99)	**0.03 ***	0.99 (0.93–1.06)	0.72
5 min Apgar score	0.88 (0.82–0.94)	**<0.001 ***	1.04 (0.94–1.14)	0.47	0.95 (0.86–1.04)	0.28	–	–
Surfactant	1.74 (1.38–2.19)	**<0.001 ***	0.81 (0.58–1.12)	0.20	0.97 (058–1.61)	0.90	–	–
Patent ductus requiring treatment	1.34 (1.03–1.73)	**0.03 ***	0.84 (0.64–1.10)	0.21	1.06 (0.79–1.40)	0.70	–	–
Intraventricular hemorrhage	2.03 (1.70–2.44)	**<0.001 ***	1.71 (1.27–2.13)	**<0.001 ***	1.56 (1.19–2.05)	**0.001 ***	0.75 (0.57–1.0)	0.05
Nutritional and anthropometrics								
Average amino acid intake	1.38 (1.02–1.88)	**0.03 ***	1.29 (0.93–1.81)	0.13	1.60 (0.98–2.59)	0.06	–	–
Average dextrose intake	0.95 (0.89–1.02)	0.16	–	–	0.91 (0.85–0.99)	**0.02**	0.96 (0.89–1.04)	0.32
Average lipid intake	0.88 (0.76–1.03)	0.11	–	–	0.99 (0.82–1.20)	0.9	–	–
Average magnesium sulfate intake	1.19 (0.16–8.81)	0.86	–	–	2.36 (0.16–33.78)	0.53	–	–
Average calcium gluconate intake	1.81 (1.28–2.54)	**0.001 ***	1.29 (0.91–1.82)	0.15	1.30 (0.83–2.02)	0.25	–	–
Average sodium chloride intake	0.64 (0.50–0.82)	**<0.001 ***	0.86 (0.68–1.09)	0.21	0.81 (0.57–1.14)	0.23	–	–
Average sodium acetate intake	0.86 (0.73–1.01)	0.07	–	–	0.82 (0.66–1.02)	0.08	–	–
Average sodium phosphate intake	0.55 (0.37–0.72)	**<0.001 ***	0.67 (0.47–0.98)	**0.03 ***	0.50 (0.32–0.78)	**0.002 ***	0.60 (0.38–0.96)	**0.03 ***
Average potassium chloride intake	0.82 (0.69–0.97)	**0.02 ***	0.98 (0.83–1.17)	0.84	0.75 (0.55–1.01)	0.06	–	–
Average potassium acetate intake	1.40 (1.17–1.67)	**<0.001 ***	1.05 (0.86–1.29)	0.61	1.17 (0.94–1.46)	0.17	–	–
Average potassium phosphate intake	1.22 (0.91–1.64)	0.18	–	–	1.23 (0.91–1.67)	0.18	–	–
Average calcium-to-phosphate ratio intake	1.13 (1.07–1.19)	**<0.001**	1.03 (0.99–1.07)	0.07	1.06 (1.02–1.10)	0.001	1.02 (0.98–1.06)	0.32

* *p*-values of <0.05.

## Data Availability

The data presented in this study are available on request from the corresponding author. The data is not available publicly due to Ethical reason.

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
