# Peer review of "Incidence and Risk Factors of Refeeding Syndrome in Preterm Infants"

_nutrients, 2024, doi:10.3390/nu16152557_

Round 1

Reviewer 1 Report

Comments and Suggestions for Authors

I have read the paper with great interest and I agree with authors that the study does have some limitations, however the strengths prevail, particularly with respect to strict definition of RFS, meticulous tracking of several parenteral nutrition components' changes and high sample size.

I also have some comments and suggestions for improvement or easier understanding.

The introduction is written very "sparsely", even for an ICU-neonatologist: the origins of RFS are insufficiently explained, particularly with the explanation why especially preterm SGA newborns can have this syndrome and why it's recognition is important (see ref. 18, Introduction or Robinson DT et al, J Perinatol 2023, Abstract).

In 2.5.1. Nutrition Protocol I am missing more precise data on nutrition composition and the guidelines for PN with respect to GA - days after birth (ref. or in individualized PN solution composition in mg/kg/min. for glucose, g/kg/day for AA, lipids, mmol/kg/day for electrolytes etc.).

Given the numbers in Figure 1 ("2465 met the inclusion criteria") I cannot understand the number "2243" in line 130; maybe some additional clarification would solve the problem. 

I find the graphics of Figure 1 - Flowchart too large (too "eye catching") - considering the limited importance of the message. Similarly, Figure 2 could be improved (narrower columns, smaller figures).

In Tables 1. and 2. I suggest to replace "Neonates aged < 28 weeks" to "Gestational age < 28 weeks". For better transparency, I also recommend graphical separation (with a line) of the columns ALL and GA < 28 weeks.

"*Statistically significant at 5% level" (lines 176, 184) is an unusual comment - perhaps use the same description of significant p-value as in line 128?

In Table 3 - Small for gestational age; explanation for Mode of delivery (cs?); Apgar 1' and 5' score; Patent ductus arteriosus requiring treatment; in Nutritional I miss the word intake at all Average ... .

In line 203, I suggest to replace "neonates aged <28 weeks" with "neonates with gestational age <28 weeks at birth".

Author Response

Reviewers' comments:

We would like to thank the reviewers for their careful reading and valuable comments on the manuscript. We have taken the comments on board to improve and clarify the manuscript. Please find below a detailed point-by-point response to all comments (reviewers’ comments in black, our replies in blue and bold).

Reviewer 1

I have read the paper with great interest and I agree with authors that the study does have some limitations, however the strengths prevail, particularly with respect to strict definition of RFS, meticulous tracking of several parenteral nutrition components' changes and high sample size.

Response: We sincerely appreciate your valuable comment.

I also have some comments and suggestions for improvement or easier understanding.

The introduction is written very "sparsely", even for an ICU-neonatologist: the origins of RFS are insufficiently explained, particularly with the explanation why especially preterm SGA newborns can have this syndrome and why it's recognition is important (see ref. 18, Introduction or Robinson DT et al, J Perinatol 2023, Abstract).

Response: We sincerely appreciate your valuable comment. As suggested we have revised and adjusted to the following “RFS was first recognized in adult patients who experienced severe malnutrition or starvation, such as prisoners of war, individuals with anorexia nervosa, or those re-covering from famine [4,5,9]. Upon refeeding, these individuals developed potentially life-threatening symptoms. The underlying mechanism involves a metabolic shift from a catabolic state, where the body primarily uses fat and protein for energy, to an anabolic state, where carbohydrate intake stimulates insulin release. Insulin promotes the cellular uptake of glucose, potassium, magnesium, and phosphate. This sudden shift can lead to profound hypophosphatemia, as phosphate is rapidly utilized in cells for ATP production and other metabolic processes [10]. The resultant electrolytes imbalance, particularly low phosphate levels, can cause severe complications, including cardiac, respiratory, and neurological dysfunction [6-8]”.

In 2.5.1. Nutrition Protocol I am missing more precise data on nutrition composition and the guidelines for PN with respect to GA - days after birth (ref. or in individualized PN solution composition in mg/kg/min. for glucose, g/kg/day for AA, lipids, mmol/kg/day for electrolytes etc.).

Response: We appreciate your novel perspective and thank you for bringing this to our attention. As suggested we have revised and adjusted to the following  “Individualized PN solution containing amino acids (3.5-4 g/kg/day), dextrose (5-12 mg/kg/min), Lipid emulsion (1-3 g/kg/day), minerals, sodium chloride (1-3 mmol/kg/day), sodium acetate (1-2 mmol/kg/day), sodium phosphate (1-2 mmol/kg/day), potassium chloride (1-3 mmol/kg/day), potassium acetate (1-2 mmol/kg/day), potassium phosphate (1-2 mmol/kg/day),trace elements (Peditrace®), and water- and fat-soluble vitamins (Soluvit® N, and Vitalipid® N Infant; respectively) was started within the first 24 h of life and infused continuously for 24 h [2]”.

Given the numbers in Figure 1 ("2465 met the inclusion criteria") I cannot understand the number "2243" in line 130; maybe some additional clarification would solve the problem. 

Response: We appreciate your keen observation. We have revised accordingly.

I find the graphics of Figure 1 - Flowchart too large (too "eye catching") - considering the limited importance of the message. Similarly, Figure 2 could be improved (narrower columns, smaller figures).

Response: As suggested, we have revised accordingly.

In Tables 1. and 2. I suggest to replace "Neonates aged < 28 weeks" to "Gestational age < 28 weeks". For better transparency, I also recommend graphical separation (with a line) of the columns ALL and GA < 28 weeks.

Response: As suggested, we have revised accordingly.

"*Statistically significant at 5% level" (lines 176, 184) is an unusual comment - perhaps use the same description of significant p-value as in line 128?

Response: As suggested, we have revised accordingly.

In Table 3 - Small for gestational age; explanation for Mode of delivery (cs?); Apgar 1' and 5' score; Patent ductus arteriosus requiring treatment; in Nutritional I miss the word intake at all Average ... .

Response: We appreciate your keen observation. As suggested, we have revised accordingly.

In line 203, I suggest to replace "neonates aged <28 weeks" with "neonates with gestational age <28 weeks at birth".

Response: As suggested, we have revised accordingly.

We thank the reviewer #1 for the constructive and insightful comments, which have helped us to substantially improve our manuscript.

Reviewer 2 Report

Comments and Suggestions for Authors

This great article examines the danger of refeeding syndromes in very early preterm infants, backed by sound statistics. The paper is clear, comprehensive and relevant. I have a remark:

Would it be possible to do a limited sub-analysis concerning extreme preterm infants ( 28 weeks and < 1,000 g)?

Comments on the Quality of English Language

Some typo's throughout the text.

Author Response

Reviewers' comments:

We would like to thank the reviewers for their careful reading and valuable comments on the manuscript. We have taken the comments on board to improve and clarify the manuscript. Please find below a detailed point-by-point response to all comments (reviewers’ comments in black, our replies in blue and bold).

Reviewer 2

This great article examines the danger of refeeding syndromes in very early preterm infants, backed by sound statistics. The paper is clear, comprehensive and relevant. I have a remark:

Response: We sincerely appreciate your valuable comment.

Would it be possible to do a limited sub-analysis concerning extreme preterm infants (≤ 28 weeks and < 1,000 g)?

Response: Thank you for your insightful comment. It is mentioned in Table 1 that we have 264 neonates less than 28 weeks, 136 babies developed RFS and their median birth weight was 775 (661.25–905) and we found that low sodium phosphate intake is a significant risk factor for RFS in this group. More details are mentioned in Tables.

Comments on the Quality of English Language

Response: We appreciate your keen observation. The certificate of editing has been attached. 

Some typo's throughout the text.

Response: We appreciate your keen observation. The certificate of editing has been attached.  

We thank the reviewer #2 for the constructive and insightful comments, which have helped us to substantially improve our manuscript.
